# Alkamides in *Zanthoxylum* Species: Phytochemical Profiles and Local Anesthetic Activities

**DOI:** 10.3390/ijms252212228

**Published:** 2024-11-14

**Authors:** I-Cheng Lu, Pin-Yang Hu, Chia-Heng Lin, Lin-Li Chang, Hung-Chen Wang, Kuang-I Cheng, Tz-Ping Gau, Kai-Wei Lin

**Affiliations:** 1Department of Anesthesiology, Kaohsiung Medical University Hospital, Kaohsiung 807377, Taiwan; u9251112@gmail.com (I.-C.L.); u8901090@gmail.com (P.-Y.H.); linsofar@yahoo.com.tw (C.-H.L.); kuaich@gmail.com (K.-I.C.); 2School of Medicine, College of Medicine, Kaohsiung Medical University, Kaohsiung 807378, Taiwan; 3Department of Microbiology and Immunology, Faculty of Medicine, College of Medicine, Kaohsiung Medical University, Kaohsiung 807378, Taiwan; m725006@kmu.edu.tw; 4Graduate Institute of Medicine, College of Medicine, Kaohsiung Medical University, Kaohsiung 807378, Taiwan; 5Department of Medical Research, Kaohsiung Medical University Hospital, Kaohsiung 807378, Taiwan; 6Department of Neurosurgery, Chang Gung Memorial Hospital, Kaohsiung Medical Center, Chang Gung University College of Medicine, Kaohsiung 833253, Taiwan; haochienwang@gmail.com

**Keywords:** *Zanthoxylum*, alkamides, phytochemistry, pharmacology, local anesthetics

## Abstract

*Zanthoxylum* species have long been utilized in traditional medicine; among their various properties, they provide an analgesic effect. Central to this medicinal application are alkamides, a class of alkaloids characterized by their unsaturated fatty acid chains. These compounds are particularly noted for their distinctive alleviation of tingling and numbing effects, which are beneficial in dental pain management and local anesthesia. This review synthesizes the existing phytochemical research on alkamides derived from 11 *Z.* species, focusing on their chemical properties, pharmacodynamics and clinical implications. The analysis includes an examination of the structure–activity relationships (SARs), pharmacokinetics and mechanisms by which these compounds modulate sensations such as pungency and numbness, contributing to their analgesic and local anesthetic efficacy. This systemic review identifies significant research gaps, including the need for comprehensive evaluations of alkamide efficacy, detailed explorations of their pharmacological mechanisms and expanded clinical applications. These areas represent key opportunities for future investigations to enhance the understanding and utilization of alkamides in medical treatments.

## 1. Introduction

Plant-derived sources have long been utilized for nutrients and medicines to treat diseases and to nourish health care in cultures worldwide. Moreover, bioactive compounds derived from plants serve as templates and provide structural designs for medicinal chemistry. Species of the genus *Zanthoxylum* L., belonging to the Rutaceae family, are distributed in tropical and subtropical regions, with the majority found in Asia, the Americas and Africa [1,2,3]. The ethnobotanical and biological reports indicate that the *Zanthoxylum* (*Z.*) species are traditionally used in managing various health conditions [4,5]. Secondary metabolites from *Z.* species extract have exhibited several pharmacological activities such as antioxidant, analgesic and anti-inflammatory activities, along with modulatory effects against obesity and diabetes [6,7,8,9]. Specifically, many species of *Z.* species distributed in tropical and subtropical regions are utilized for pain relief in relation to toothaches and post-operative pain, including *Z. nitidum* (Roxb.) DC. [10], *Z. zanthoxyloides* Lam [11,12,13,14], *Z. schinifolium* [15], *Z. elephantiasis* [16], *Z. riedelianum* Engl. [17], *Z. Americanum* Mill. [18], *Z. bungeanum* Maxim. [19], *Z. gilletii* (De Wild) P.G. Waterman [20], *Z. chalybeum* Engl. [21], *Z. oxyphyllum* Edgew. [22] and *Z. heitzii* (Aubrév. and Pellegr.) P.G. Waterman [23]. Those studies, mentioning pain attenuation in traditional uses, indicate that *Z.* species can be associated with potential sources for the drug development of local anesthetics.

The phytochemical profile of *Z.* species is diverse, encompassing terpenes, flavonoids, coumarins, phenolic acids and alkaloids. [4,5]. Among the alkaloids, alkamides represent a special class of bioactive constituents, characterized by straight-chain, unsaturated fatty acids linked via an amide bond to various amines, forming distinctive natural products [24,25]. Previous studies have revealed that alkamides are mostly found in the *Z.* species with, consequently, the traditional use of such *Z.* species herbs due to being rich in alkamides [26,27,28]. Alkamides exhibit a broad spectrum of biological activities, including antifungal, antibacterial, immunomodulatory, antidiabetic, anti-inflammatory, analgesic and neuroprotective effects. [29,30,31,32,33,34,35]. The unique tingling and numbing sensations in *Z.* species are attributed to the plants containing a collection of alkamides [25]. Furthermore, alkamide extracts or whole plants rich in alkamides are used in the treatment of toothaches or as anesthetics [36,37]. Previous reviews have shown that a series of alkamides in the genus *Z*. exhibit numbing and tingling sensations in sensory neurons [38], while a series of reviews reported that the alkamides from *Z* species or other plants caused numbing sensations and analgesic effects [24,28,39].

This review aims at providing a systematic and updated analysis of alkamides from 11 *Z.* species, focusing on their phytochemistry, pharmacological actions, pharmacokinetics and clinical applications with particular emphasis focused on the potential of these compounds as local anesthetics in the process of examining their sensory and analgesic effects. By synthesizing current knowledge and identifying research gaps, this review seeks to guide future investigations into the development of alkamides as potential local anesthetics.

## 2. Phytochemistry

### 2.1. Chemical Properties and Metabolism in the Plants

Alkamides are widely distributed in plants and are structurally related to *N*-acyl-L-homoserine lactones (AHLs) found in Gram-negative bacteria, as well as to *N*-acylethanolamines (NAEs) found in plants and mammals [40]. Accordingly, these compounds contain a fatty acid tail, which comprises saturated or unsaturated carbon chains with lengths ranging from C_8_ to C_18_ combined with an amide group and a variable headgroup. The olefinic double bonds of alkamides display *E* or *Z* forms depending on whether the higher priority groups (methylene) on both carbons of the double bond are on opposite sides or the same side, respectively. The characteristics of *E*/*Z* isomers, including the level of saturation, number of carbon atoms and stereochemistry, serve as chemical taxonomic criteria for different plant families [40].

The chemical structure of the alkamide family is unstable and sensitive to oxygen, light and heat due to the presence of unsaturated fatty acids containing olefinic bonds [41]; for example, the content of sanshool alkamides in an aqueous solution at pH 7.0 was reduced by 50% at room temperature over four weeks. These compounds are also susceptible to degrade at high temperatures, and an 80% ethanolic solution containing 200 ppm of alkamide exposed to ultraviolet (UV) light completely disappeared within four hours [42]. The double-bonded part of the alkamides might undergo changes under ultraviolet B (UVB) irradiation, leading to the interconversion of chemical structures [42]. A previous study tried to evaluate the degradation of alkamides in plant extracts under different storage conditions. Alkamides in DMSO solution degrade more slowly than in dry films, and, if alkamides are combined with phenolic acids acting as antioxidants in dry films, the degradation appears slower than that of alkamides with phenolic acids in DMSO solution [43]. Based on the above studies, this indicates that pure alkamides should be kept in solvents (except for acidic solutions). Low temperature and protection from light irradiation are essential for the storage of pure alkamides.

The alkamides produced in plants are usually responsive to environmental stress and promote plant growth. In plants, alkamides are secondary metabolites that exhibit a chemical defense against microbial and herbivorous predators [44]. The related metabolic pathways, dependent on phytohormones, lead to the expression of defense-related genes and the production of antimicrobial secondary metabolites [45]. Studies on alkamide generation and accumulation in *Z.* species have demonstrated that ZbFAD2 and ZbFAD3 are vital alkamide biosynthesis enzymes in *Z. bungeanum* via the promotion and regulation of the production of alkamides [46]. In plant development, alkamides produce unique stimulatory effects to control differentiation processes during plant growth through interaction with the cytokines in the signaling pathway [47]. Alkamides can also regulate NO production to promote lateral and adventitious root formation at various stages of *Arabidopsis* explant development [48].

### 2.2. Alkamides Extraction, Isolation and Structure Elucidation

Extraction is the initial stage in the process of isolating compounds from plant materials, and various methods of plant extraction can be tailored to target specific phytoconstituents and structures. The temperature and physicochemical conditions of extraction methods, which include maceration, infusion, decoction, Soxhlet extraction, counter-current extraction, sonication, supercritical fluid extraction, hydrodistillation, microwave-assisted extraction and ultrasound-assisted extraction, are chosen based on the constituents and properties of the plants involved [49,50].

Various solvent systems such as hexane, chloroform, ethyl acetate, ethanol and methanol are commonly employed for extracting alkamides from plant materials. Among these, chloroform is the most suitable solvent system, and, although both methanol and ethanol have also been used [40], studies have shown that water, methanol and ethanol extracts of *Z. bungeanum* tend to contain higher concentrations of alkamides compared to extracts obtained using other solvent systems. These three solvent extracts in particular are noted for their significant content of major alkamides such as ZP-amide C and ZP-amide D, suggesting that water, methanol and ethanol are effective solvents for extracting these specific alkamides from *Z. bungeanum* [51], with two previous studies demonstrating that ZP-amide C and ZP-amide D were isolated from the methanol extract of *Z. piperitum* [52], while 70% ethanol extract was used as solvent to extract and separate the alkamides from *Z. bungeanum* pericarps [53].

Chemical detection and identification of the structural characterization of compounds is a significant challenge in natural product research. Various solid-phase chromatography techniques play a crucial role in separating and analyzing these compounds. Coating materials used in chromatography assays such as thin-layer chromatography (TLC), column chromatography (CC), flash chromatography (FC), size-exclusion chromatography (SEC), medium-pressure liquid chromatography (MPLC) and high-performance liquid chromatography (HPLC) have all been successfully employed for separating natural products with similar structures [54].

Isolating and purifying alkamides from plants poses challenges primarily due to their structural similarity and tendency to exist as racemic mixtures. The structural similarity among alkamides makes it difficult to separate them using conventional purification methods; additionally, their racemic nature further complicates the isolation process because enantiomers (mirror-image forms) have identical physical and chemical properties, making it challenging to distinguish and separate them [24,55]. To overcome these challenges, chromatography-coated materials suited to a special structure are necessary, such as chiral chromatography, which can separate enantiomers based on their stereochemistry. For *Z. bungeanum*, two alkamide enantiomers were successfully isolated by preparative HPLC with a chiral column from a racemic mixture [56]. In order to confirm the structure of pure compounds obtained through the separation process, spectroscopic analysis, including Fourier-transform infrared spectroscopy (FTIR), Ultraviolet–visible spectroscopy (UV–Vis), nuclear magnetic resonance spectroscopy (NMR) and mass spectrometry (MS) are used for structural elucidation. The circular dichroism (CD) spectrum combined with calculated electronic circular dichroism (ECD) and optical rotation are applied for absolute configurations of alkamides.

To date, about 65 alkamides have been isolated and identified from eleven *Z.* species. This review presents the phytochemical studies of alkamides obtained from 1957 to 2022. Their structures are shown in Figure 1, Figure 2, Figure 3 and Figure 4.

#### 2.2.1. Alkamides from *Zanthoxylum piperitum* DC

The α-sanshool (**1**) isolated from a light petroleum extract of the ground bark of *Z. piperitum* through column chromatography on neutral alumina was the first alkamide obtained from *Z.* species [57], while Hydroxy-α-sanshool (**2**), β-sanshool (**3**) and (2*E*,4*E*,8*E*,10*E*,12*E*)-*N*-Isobutyl-2,4,8,10,12-tetradecapentaenamide (named **δ**-sanshool) (**4**) were isolated from a CHCl_3_ extract of the bark of *Z. piperitum* through normal-phase HPLC [58,59]. Hydroxy-ε-sanshool (**5**) was isolated from the seed of *Z. piperitum* by C_18_ reversed-phase HPLC using 30% aqueous MeOH [60], where **2** and **5** revealed a hydroxyl group at isobutyl C2 position [58,60]. Six alkamides, ZP-amide A (**6**), B (**7**), C (**8**), D (**9**), E (**10**) and F (**11**) with unsaturated fatty acid amides with hydroxyl and carbonyl substitutions were purified from the pericarp of *Z. piperitum* fruits, and, among these, **8** and **9** were racemic mixtures owing to optical inactivity. The relative configuration of **8** and **9** at C-10 and C-11 remains to be analyzed, while **11** is the stereoisomer of **10** with the relative configuration at the two asymmetric carbons, C-6 and C-11, **10** and **11** also being racemic mixtures owing to optical inactivity. The relative configuration and optical inactivity results of **8**–**11** were determined by 1D and 2D NMR and optical rotation [52]. Hydroxy-ζ-sanshool (**12**) was isolated from supercritical fluid extract in MeOH through MPLC and reversed-phase HPLC, where the two *cis*-configuration of double bonds in **12** was confirmed by expected cis-shielding effect of carbon atoms C-7 and C-14; finally, the *trans*-configuration of the double bond in **12** was detected by coupling constant of the two double duplets detected for H-10 and H-11 [61].

#### 2.2.2. Alkamides from *Zanthoxylum bungeanum*

The CHCl_3_ extract of *Z. bungeanum* was chromatographed on a silica gel column and further isolated with reversed-phase HPLC to derive hydroxy-β-sanshool (**13**), (2*E*,4*E*,8*E*,10*E*,12*E*)-2′-Hydroxy-*N*-isobutyl-2,4,8,10,12-tetradecatetraenamide (named hydroxy-γ-isosanshool) (**14**) and (2*E*,4*E*,8*Z*,11*Z*)-*N*-(2-Hydroxy-2-methylpropyl)-2,4,8,11-tetradeeatetraenamide (named bungeanool) (**15**). These three alkamides contained a hydroxyl substitution at isobutyl C2 position [62]. A double-bond moiety linking to NH through a methylene group named dehydro-γ-sanshool (**16**) was isolated from an EtOH extract of dried pericarps of *Z. bungeanum*, while tetrahydrobungeanool (**17**), dihydrobungeanool (**18**) and isobungeanool (**19**) were also isolated from an EtOH extract of dried pericarps of *Z. bungeanum* [63]. (2*E*,7*E*,9*E*)-*N*-(2-hydroxy-2-methylpropyl)-6,11-dioxo-2,7,9-dodecatrienamide (**20**) containing two ketones located on an unsaturated carbon chain was isolated from a MeOH extract of pericarps of *Z. bungeanum*. In particular, (2*E*,6*E*,8*E*)-*N*-(2-Hydroxy-2-methylpropyl)-10-oxo-2,6,8-decatrienamide (**21**) containing a terminal aldehyde group on the unsaturated side chain was also isolated from the same extract [64].

Recent studies indicate that some alkamides have been successfully isolated from an EtOH/water plant extract. The pericarps of *Z. bungeanum* underwent treatment via 70% EtOH that was purified with a Sephadex LH-20 and preparative reversed-phase HPLC system to produce qinbunamide A–C (**22**–**24**), where **22** and **23** contained an ethoxy moiety attached on C-6 and C-11, respectively, while **24** contained carbonyl and hydroxyl groups on C-6 and C-11, respectively. Olefinic, carbonyl, hydroxyl and ethoxy groups in positions **22**–**24** were determined by 2D NMR; however, the stereochemistry of the ethoxy and hydroxyl moiety of **22**–**24** was still evidenced as a relative configuration [53]. A series of isobutylhydroxyamides **25**–**32** containing unsaturated carbon chains with carbonyl and hydroxyl groups were isolated from a 95% ethanol extract of the pericarp of *Z. bungeanum* [56].

In this study, **6** and **7**, as obtained in a previous study [52], were separated by chiral columns to obtain enantiomers **6a**/**6b** and enantiomers **7a**/**7b**, respectively. The absolute configuration of enantiomeric **6a** and **7a** were determined as 6*R* and 11*R*, respectively; therefore, **6b** and **7b** were determined to be 6*S* and 11*S*, as they are the enantiomers of **6a** and **7a**. Compound **30** was a stereoisomer of **29** with different relative configurations at C-6 and C-7; **29** and **30** were also confirmed as *erythro* and *threo* forms, respectively. The relative configuration of **32** was the *erythro* form due to the coupling constant between H-6 and H-7 being the same as **29** [56]. Zanthoamide A (**33**), B (**34**), C (**35**) and D (**36**) and bugeanumamide A (**37**) were isolated from a 95% EtOH extract of powdered pericarps of *Z. bungeanum*. In addition, **33**–**35** also exhibited an unsaturated carbon chain substituted by carbonyl and hydroxyl groups. These four alkamides were racemic mixtures on account of the optical inactivity of these compounds. Compound **37** contained a rare C-6 fatty acid with an acetal group confirmed by 2D NMR correlation of OCH_3_ with H-6 and six protons of the two methoxyl groups as a strong singlet in ^1^H NMR [65].

#### 2.2.3. Alkamides from *Zanthoxylum lemairie* (De Wild.)

Two aromatic alkamides, **38** and **39**, were isolated from a CHCI_3_ extract of pericarps of *Z. lemairie*, revealing that the aromatic moiety was linked to the methoxy groups. Compared with the spectroscopic data of **38** and **39**, the NMR absorption of *N*-CH_3_ on **39** confirmed that **39** is the *N*-methyl derivative of **38** [66].

#### 2.2.4. Alkamides from *Zanthoxylum Integrifoliolum* (Merr.) Merr

Lanyuamides I–III (**40**–**42**) were isolated from a MeOH extract of fruits of *Z*. *Integrifoliolum*. The carbonylation on C-8 and C-12 of **40** and **41**, respectively, were determined by the difference of the ^1^H NMR spectrum of methylene groups neighboring the keto group [67].

#### 2.2.5. Alkamides from *Zanthoxylum ailanthoides* (Sieb. et Zucc.)

γ-sanshool (**43**) and hydroxy-γ-sanshool (**44**) were isolated from a MeOH extract of *Z. ailanthoides*. In the ^13^C NMR spectrum of **44**, the carbon signal attached to hydroxyl moiety was shifted downfield in comparison with that of **43** and singlet in the off-resonance ^13^C NMR spectrum [68]. (2*E*,4*E*)-*N*-isobutyl-6-oxohepta-2,4-dienamide (**45**) was isolated from a MeOH extract of the stem bark of *Z. ailanthoides* through being chromatographed on silica gel and preparative TLC. The terminal acetyl group of the fatty acid of **45** was elucidated by 1D and 2D NMR correlation of C-4 to C-7 [69].

#### 2.2.6. Alkamides from *Zanthoxylum armatum* DC

Dried pericarps of *Z. armatum* were extracted in MeOH to educe four alkamides, timuramides A–D (**46**–**49**), which were purified with diol batch elution, LH-20 Sephadex chromatography and C_18_-HPLC. In order to confirm absolute configuration of the endoperoxide ring of **46**, it was converted into a secondary diol (**46a**) by hydrogenolysis of the peroxide with Lindlar’s catalyst [70]. The 1D NMR of **47** was like **46**, except that it contained a C-6/C-7 trans-olefinic bond in contrast to the cis-configured bond in **46**. The 1D NMR of **48** was similar to **46** and **47**, except that it lacked an endoperoxide moiety and a terminal carboxylic acid at C-10 of **48**. The 1D NMR of **49** was similar to **48**, except that it lacked both C-6/C-7 and C-8/C-9 pairs of olefinic carbons [71].

#### 2.2.7. Alkamides from *Zanthoxylum nitidum* (Roxb.) DC

Four unsaturated alkylamides, *Zanthoxylum amides* A–D (**50**–**53**), were purified from a 60% EtOH extract of the roots of *Z. nitidum*, using silica gel column CC by gradient elution and semi-preparative HPLC. The absolute configuration of hydroxylation and carbonylation carbons on the unsaturated structure of **50**–**53** could not be established due to the small amount of residue [72]. *Zanthoxylum amides* J–P (**54**–**60**) were isolated from the whole plant of *Z. nitidum*, extracted with 95% EtOH. The purification process included silica gel column CC and a Sephadex LH-20 column followed by reversed-phase HPLC. Compounds **54**–**60** exhibited hydroxyl, ethoxyl and carbonyl substitutions at an unsaturated carbon chain. The final configurations of racemic mixtures including **54** and **57**–**60** were determined by ECD [73]. (2*E*,6*E*,8*E*)-*N*-(2-methylpropyl)-10-oxo-2,6,8-decatrienamide (**61**) was isolated from the stems of *Z. nitidum* extracted with CH_2_Cl_2_. The unsaturated alkylamide of **61** possessing an aldehyde group was determined by 1D and 2D NMR spectrum, including the 2D NMR correlations between olefinic protons and aldehyde carbonyl carbon [74].

#### 2.2.8. Alkamides from *Zanthoxylum nitidum* var. Tomentosum

From the dried whole herb of *Z. nitidum*, *Zanthoxylum amides* E–I (**62**–**66**) were purified by chromatography over silica gel, a Sephadex LH-20 column (MeOH) and semipreparative HPLC. The absolute configurations of the hydroxyl and methoxyl carbons of chiral racemic isobutylamides **62**–**65** were determined by ECD and 1D and 2D NMR, while the positions of enol moieties on the unsaturated structure of **62**–**66** were also confirmed by NMR spectrum [75].

#### 2.2.9. Alkamides from *Zanthoxylum heitzii*

Pellitorine (**67**) and 6-hydroxypellitorine (**68**) were isolated from the bark of *Z*. *heitzii* extracted with hexane in a Soxhlet. The purification process was through the CH_2_Cl_2_/EtOAc system by silica gel column CC [76]. Compound **67** contained a secondary amide with an α,β-unsaturated conjugated system [77], while **68** was the derivative of C-6 hydroxylation of the unsaturated structure [78].

#### 2.2.10. Alkamides from *Zanthoxylum zanthoxyloides*

Zanthoamides G (**69**) and I (**70**) were isolated from the fruits of *Z. zanthoxyloides* by reversed-phase preparative HPLC. Compounds **69** and **70** showed the hydroxylation of C-13, and **69** was a racemic mixture containing dihydroxyl moieties on C-12 and C-13, with a ketone carbonyl moiety being present on C-8 of **70** [79].

#### 2.2.11. Alkamides from *Zanthoxylum chalybeum*

4-(isoprenyloxy)-3-methoxy-3,4-deoxymethylenedioxyfagaramide (**71**) was isolated from the ground stem bark of *Z. chalybeum* and extracted with sufficient volumes of 50% MeOH in CH_2_Cl_2_, containing an aromatic moiety attached to isoprenyloxy and methoxy groups [80].

## 3. Pharmacology: Local Anesthetic and Analgesic Effect

A local anesthetic provides its analgesic effect by inhibiting pain transmit pathways via a blockade of type A delta or C nerve impulses on sensory nerves [81]. The main anesthetic activities in the nervous system act on voltage-gated sodium channels (Nav), which are blocked and modulated to prevent conduction, thereby inhibiting sensory nerves’ communication [82]. To evaluate the main constituents of herbs possessing the ability to blockade sensory nerve transmission, the local anesthetic characteristics should be carefully identified. Sometimes, local analgesic properties may be mistaken to show local anesthetic effect but present a dysfunctional sensory nerve through the inhibition of transient receptor potential (TRP) channels [83]. The extracts from Z. species have been demonstrated to modulate sensory functions, such as suppressing pain [8,9]. Therefore, the Z. species’ pharmacological properties may have the potential to identify its main constituents closely associated with local anesthetic effects.

In this section, alkamides extracted from Z. species possessed pharmacological properties ranging from pungent and tingling sensations to analgesic and to local anesthetic effects (Table 1). In relation to alkamides’ regulation of main bioactivities and drug development perspectives, the structure–activity relationship of alkamides for their pharmacologic properties is also explored in further detail.

### 3.1. Pungent and Tingling Properties

Alkamides exhibiting pungent and tingling properties have been reported in species of *Z*. of the Rutaceae family. Compounds **2** and **5** produce similar tingling sensations when applied to the human tongue. Compound **2** has been observed to induce numbness during toothache treatment. In tests for spontaneous activity of the sensory system, compound **2** caused insensitivity to innocuous thermal or tactile stimuli as well as insensitivity to touch or cooling [85]. In a sensory functional evaluation of the alkamides **1**–**4**, **13** and **43** from *Z. bungeanum*, compound **1** exhibited burning and tingling qualities and lasted the longest period in terms of stimulus duration among all sanshool compounds. Compound **3** was perceived as numbing and bitter, compound **4** as burning, numbing and fresh, compound **43** as burning and fresh, compound **2** as tingling and numbing, and compound **13** as numbing, astringent and bitter [84]. In the study reported by Bader et al., the sensory assay for evaluating pungent and/or tingling sensations showed that sanshool derivatives containing a *Z*-configuration double bond such as **2**, **5**, **15**, **19** and **44**, exhibited a tingling and paresthetic sensation; however, derivatives with an all-*E* configuration, specifically **13** and **14**, induced numbing and anesthetic effects [53]. Compound **67**, which is also an all-*E* configured alkamide, produces a numbing sensation on the tongue at a concentration of 10 ppm [90].

### 3.2. Analgesic Effect

The sensation is modulated by some key factors reported in previous studies. Transient receptor potential (TRP) channels are responsible for the sensations, which are thought to mediate pain and respond to natural compounds [91]. Compound **2** promoted Ca^+2^ influx in cells transfected with TRPV1 or TRPA1 and evoked robust inward currents in dorsal root ganglia neurons and trigeminal ganglion neurons transfected with TRPV1 or TRPA1 [86]. Compound **43** was a potent agonist of TRPV1 activity, which explains its pungent and tingling sensation, as well as its use as a natural anesthetic for toothache [84]. In electrophysiological experiments using *Xenopus* oocytes expressing compound **2**, sensory neurons were excited through the inhibition of pH- and anesthetic-sensitive two-pore potassium channels (KCNK3, KCNK9 and KCNK18). Activation of TRPA1 and TRPV1 channels is mediated by sanshool alkamide **2**, which blocks outward K^+^ current by activating KCNK channels. These channels are also targeted by volatile anesthetics, supporting the traditional use of *Z*. species extracts in folk medicine for treating toothache [87]. In behavioral assays, fiber recordings, calcium imaging and whole-cell electrophysiology of cultured sensory neurons, compound **2** inhibited the activity of multiple voltage-gated sodium channel subtypes, with Nav1.7 being the most strongly affected. It also inhibited Aδ mechanonociceptors, which mediate both sharp acute pain and inflammatory pain [88].

### 3.3. Local Anesthetic Effect

In the formalin test used to evaluate anesthetic effects, nanostructured lipid carriers loaded with compound **2** exhibited a significant anesthetic effect at a low dose compared to free compound **2** and the positive control lidocaine. Alkamides like compound **2** demonstrated potential for local analgesic effects. Regarding the duration of local anesthesia, formulation with compound **2** resulted in rapid onset and a sustained longer effect [89]. Until now, there have been few studies investigating the local anesthetic effects of pure alkamides, including in vivo and behavioral assays. Li et al. reported that water extracts from different processed products (high and low dosage groups) of *Z. bungeanum* exhibited a local anesthetic effect on the isolated sciatic nerve of rats [92]. In the Ma et al. report, water extract of *Z. bungeanum* exhibited higher alkamide contents compared to other solvent systems [51], indicating that some alkamides in the extract with local anesthetic effect are still unknown. In particular, plant extracts that are rich in alkamides are used in treatment as anesthetics in folk medicine [40]. It is necessary to determine the known or unknown alkamides isolated from plants and to further identify their local anesthetic effects.

This section explores the inference process from sensory properties such as numbing, modulation by key proteins including the TRP and KCNK series, and modulation factors for analgesic effects as determined from local anesthetic effects observed in assays. The above results indicate that alkamides have potential in the development of local anesthesia.

### 3.4. Structure–Activity Relationship for Pharmacology

Alkamides contain acyl chains with E/Z configurations double bonds. According to the sensory properties reported by Sugai et al., sanshool alkamides containing 2*E* and 6*Z*-double bonds in the acyl chain exhibited tingling sensations, similar to compounds **1** and **2**. Alkamides with 2*E-* and 4*E*-double bonds in the acyl chain, such as compounds **4** and **43**, were perceived as having a fresh sensation [84]. The stimulus duration assay indicated sanshool alkamides containing a *Z*-double bond, like **1**, **2** and **43**, exhibited longer stimulus than all-*E* isomers like **3**, **4** and **13**. Compound **1** also lasted the longest time in the measurement for duration of the stimulus [84]. It appears that 2*E* and 6*Z*-double bonds in the acyl chain are necessary for tingling intensity; furthermore, the *Z*-double bond in the acyl chain seems to be a critical element for sensory properties. According to the report by Bader et al. on the perception of pungency in hydroxy sanshool derivatives, compounds **2**, **5**, **15**, **19** and **44** containing the Z-configuration exhibited a tingling sensation. In contrast, compounds **13** and **14** containing the all-*E* configuration exhibited a numbing and anesthetic sensation [61]. However, Sugai et al. reported in 2005 that compounds **1**, **2** and **43** containing a *Z*-double bond also exhibited a numbing sensation [85].

As previously mentioned, the E/Z configuration possesses pungent and tingling properties. Other properties, like carbon chain length, hydroxylation on an isopropyl group and pattern of unsaturation, are also involved in sensory-active recognition. In the previous study, Galopin and Furrer tried to synthesize various sanshool alkamides and then explore the relationship between the structural composition of the numbing substances, with the numbing generation of proposed structural requirements shown in Figure 5. The minimal requirement for pungent intensity was R = H, *n* = 1, x = 1, and no conjugated *Z*-double bond existed in the structure. The additional optional feature could generate an obvious numbing sensation, i.e., two of the three conditions of the optional feature (R = OH, *n* = 2, x > 2) replacing the minimal requirements are sufficient to generate obvious numbing [93]. However, when the R group is replaced with -OH, as seen in compounds **2**, **5**, **15**, **19** and **44**, they exhibit tingling and paresthetic sensations, rather than numbing. [53]. On the other hand, sensory evaluations of a series of diols alkamides, including **8** and **9** from *Z. piperitum* and **54** and **65** from *Z. nitidum*, are still unknown. According to the hypothesis of structure in the Galopin and Furrer report (Figure 5), the structure of **65** may generate numbing sensation (R = H, *n* = 1, x > 2), while compounds **8**, **9** and **54**, which possess two optional features (**8** and **9**: R = OH, *n* = 1, x > 2; 54: R = H, *n* = 2, x > 2) might generate a more pronounced numbing sensation compared to compound **65** [52,75,93]. Compound **28** with an enol, **45** with an endoperoxide ring and **48** with a carboxylic acid moiety could also induce a more obvious numbing sensation (**28**: R = OH, *n* = 2, x > 2; **45** and **48**: R = OH, *n* = 1, x > 2) [52,69,93]; however, the relationship between an oxygenated unsaturated carbon chain and the numbing sensation remains unclear. Accordingly, the pungent properties, including the numbing sensation of alkamides with oxygenated unsaturated carbon chains, should be further evaluated.

Previous studies have demonstrated that the *Z* configurations for double bond on the aryl chain of sanshool alkamides was an essential property for TRPV1 channel activation. A series of synthetic alkamides containing the modification of the unsaturated alkyl moiety from the *Z*-olefins structure was crucial in the activation of TRPA1 receptors [94], indicating that the 5Z or 6Z position of the double bond of alkamides is necessary for activating the TRPA1 channel (Figure 5), although the evaluation of TRPA1 and TRPV1 activities has only focused on compound **2** and its synthetic derivatives. To accurately confirm the active structure, more alkamide derivatives similar to compound **2** should be evaluated for their activities on TRPA1 and TRPV1 channels. In the Koo et al. report, compound **2** with 6*Z*-double bond activated TRPA1 and TRPV1 by evoking Ca^2+^ influx in cells, while, in the small sensory neurons, TRPV1 and TRPA1 were expressed primarily to mediate pain [86]. According to the hypothesis that the 6*Z*-double bond configuration in compound **2** is necessary for the activation of the TRPV1 and TRPA1 channel, a series of analogs such as compounds **5**, **8**, **9**, **45**, **48** and **65** also containing the 6*Z*-double bond might activate TRPV1 and TRPA1 to regulate pain.

The pharmacological study of alkamides has only focused on compound **2** for its local anesthetic effect, although the water extract of *Z. bungeanum* exhibited a significant local anesthetic effect on the isolated sciatic nerve of rats [92], where the compounds **8** and **9** were major alkamides found in the water extract of this variant. These two alkamides possess a 6*Z*-double bond and a hydroxyl group on the isopropyl moiety [51]. According to the Galopin and Furrer report, the structures of **8** and **9** could generate a more obvious numbing sensation (Figure 5) [52,93]. This biological property is related to the local anesthetic effect. The above results indicate that **8** and **9** might be the key constituents for the local anesthetic effect, and these two alkamides have the potential to be evaluated for their local anesthetic effect through detailed activity screening and extensive animal studies.

## 4. Pharmacokinetics

The pharmaceutical-grade traditional Japanese medicine, daikenchuto (TJ-100), was investigated in healthy Japanese volunteers after a single oral administration. Compound **2** as an ingredient in TJ-100 reached its maximum plasma concentration within 30 min after administration, with a median half-life of 1.6 to 1.7 h. The maximum plasma concentration of **2** ranged from 0.76 to 2.66 µM [95].

The chromatography method has also been applied for pharmacokinetic study effectively. A sensitive UHPLC–MS/MS method was developed and validated for determination of alkamides **2**, **13** and **44** in rat plasma after subcutaneous and intravenous injections of *Z. bungeanum* EtOH extract. The results of plasma concentration indicated that **2**, **13** and **44** were rapidly absorbed after subcutaneous administration, and all achieved a *C*_max_ within 1 h, with the elimination half-life (*t*_1/2_) of these three alkamides being no more than 2.5 h. The subcutaneous absolute bioavailability of **2**, **13** and **44** were 100.2, 76.2 and 90.3%, respectively. These results indicate that **2** was wider and more rapidly distributed in the plasma compared with **13** and **44** due to its all-*trans* polyene structure with a lower polarity and a higher oil distribution coefficient [96].

The pharmacokinetic analysis of compound **38** in the brain showed that it could be absorbed into the blood and rapidly cross specialized barriers like the blood–brain barrier (BBB) after oral administration. It is metabolized in the liver and excreted via the kidneys. Compound **38** was quickly distributed to brain tissue (within 5 min post oral administration) and was finally excreted approximately 4 h post oral administration [97]. The lipid-soluble structure of alkamides enhances their permeability through the blood–brain barrier (BBB). Current studies have indicated that alkamides from *Z*. species exhibit favorable absorption, elimination in the bloodstream and penetration into brain tissue, although these studies have primarily focused on a limited number of alkamides such as **2** and **38**, highlighting the need for more comprehensive pharmacokinetic analyses across various types of alkamide.

Drug metabolism is also a crucial aspect of pharmacokinetics, influencing predictions of drug efficacy, safety and toxicity. Sanshool-type alkamides, such as compound **2**, are the main active components in *Z. bungeanum* and have been the focus of drug metabolism research. A recent study indicated that the metabolic stability of **2** in human and rat liver microsomes, as well as human hepatocytes, was superior to that in monkeys, dogs and mice. Compound **2** also demonstrated strong inhibitory effects on CYP2C9 and CYP2D6 in human liver microsomes [98]. In *Z*. species, alkamides exhibit metabolic stability and strong non-specific binding to plasma proteins, although the detailed molecular mechanisms, including protein-catalyzed metabolite production, remain unclear. Previous studies have indicated that the synthetic isobutylamide (2*E*,4*E*,8*Z*,10*Z*)-*N*-isobutyldodeca-2,4,8,10-tetraenamide undergoes epoxidation at double bonds catalyzed by microsomal cytochrome P450 enzymes. Metabolites of synthetic isobutylamide detected by LC-MS showed an apparent molecular weight of 263 amu, suggesting simple epoxidation or hydroxylation [99]. Subsequent studies have revealed that different forms of P450 exhibit specific catalytic reactions. Two isoforms, CYP1A1 and CYP1A2, produced identical epoxide and *N*-dealkylation products, while, in contrast, CYP2D6 generated two distinct epoxides and a hydroxylated metabolite [100]. The above studies infer a relationship with the metabolism of alkamides in *Z.* species, allowing for predictions on how enzymes catalyze the formation of epoxidation or hydroxylation metabolites. According to this inference, the *Z* form double bonds in the structure are most likely epoxidated or hydroxylated by P450.

## 5. Clinical Application

The pharmacokinetic study of the alkamides of plants allows the prediction of clinical usage. In human volunteers, the plasma concentration results of two alkamides in the traditional Japanese medicine TJ-100, including the maximum plasma concentration after administration and the elimination half-life, provide crucial information for their clinical application. These results also indicate how to carefully control the administration range to prevent excessive absorption and the potential danger of side effects [95]. In the Rong er al. report, the pharmacokinetic study of the alkamides of *Z. bungeanum* EtOH extract, including plasma concentration, elimination half-life and subcutaneous absolute bioavailability, provides valuable information for improving the prediction of clinical outcomes. The pharmacokinetic behaviors of **2**, **13** and **44** in *Z. bungeanum* EtOH extract were used for the clinical application of anesthetics. To prevent potential side effects and ensure better clinical application, the dosage of compound **2** was carefully and accurately controlled due to its high subcutaneous bioavailability [96].

The formalin test is utilized in preclinical pain research to elucidate the action mechanisms and analgesic effects of novel compounds, particularly those related to their local anesthetic effects [97]. Nociceptive behavior triggered in the formalin test consists of two phases: in phase I, the nociceptors are activated by formalin acutely through non-nociceptive Aβ fibers; in phase II, the activity is driven partly by central sensitization of spinal cord circuits [101]. In electrophysiological studies, Aδ- and C-fiber nociceptors exhibit activity during both phases of the formalin test [102], and, in this test, compound **2** combined with nanostructured lipid carriers revealed remarkable pain-relieving effects at various phases of the formalin test compared to the control group and lidocaine. Compound **2** probably could inhibit nociceptors (such as C-fiber, Aβ-fiber and Aδ-fiber) in the somatosensory neurons through blocking various Nav channel subtypes to reduce pain behavior [89]; accordingly, compound **2** might serve as a potential local anesthetic in clinical applications.

## 6. Conclusions and Perspectives

Alkamides that are present and enriched in numerous *Z*. species of plants are recognized for their pain-suppressing properties and their potential in the development of local anesthetic research. Various traditional medical practices involving these plants have shown beneficial effects in pain modulation globally. Current phytochemical studies indicate that alkamides exhibit specific stereochemical characteristics due to isomerization of their configurations, often existing as racemic mixtures in plants despite their relatively simple and similar structures; consequently, the detection, isolation and elucidation of a series of alkamides using various chromatographic and spectroscopic methods are crucial for confirming the constituents present in these plants.

The research on pungent, tingling and numbing sensations, as well as the analgesic effects of alkamides, is crucial for understanding their local anesthetic properties. These properties are intricately linked to their structural characteristics and are essential considerations in drug design, although current studies on local anesthetic effects and related bioactivities of alkamides are limited, particularly regarding specific compounds, and few studies have deeply investigated the mechanisms underlying their local anesthetic effects and related analgesic properties. Thus, key questions remain that need to be answered in future research:(a)Until now, the efficacy of compound **2** has been demonstrated in relatively complete studies, including its pungent and tingling properties, the mechanism of analgesic effect and the local anesthetic effect; however, no other alkamide has been identified with similarly detailed molecular mechanisms for these effects and properties, and a series of alkamides similar to **2** should be further investigated accordingly. The examination of alkamides for local anesthetic effect and related bioactive properties should be wide-ranging and not merely focused on specific compounds.(b)The studies on alkamides should not be limited to phytochemistry and primary in vitro assays but should encompass a full series of investigations of their pharmacological effects.(c)The SARs of alkamides related to their pungent, tingling, analgesic and local anesthetic effects have been hypothesized and remain largely theoretical, requiring detailed investigations to confirm these relationships.(d)The results of pharmacokinetic studies are pivotal for clinical applications, as they furnish essential references for subsequent clinical research. Despite this significance, only a limited number of alkamides or crude extracts enriched with alkamides have been directly tested in clinical trials. It is therefore crucial to emphasize the need for a broader exploration and evaluation methodology of various types of alkamides through clinical testing, which is essential to substantiate their safety and efficacy profiles, thereby facilitating their correspondently safe and effective use in clinical settings.

In future studies, it is imperative to deepen research on alkamides or plants containing alkamides, focusing on the issues outlined above, to advance the development of local anesthesia from natural sources. This review aims to consolidate the current studies on phytochemistry and pharmacology related to local anesthetic effects and other activities, pharmacokinetics and clinical applications of alkamides derived from *Z*. species. Such an approach could accelerate the development of natural, plant-derived local anesthetics from Z. species that could potentially offer safer and more effective alternatives to their synthetic counterparts.

## Figures and Tables

**Figure 1 ijms-25-12228-f001:**
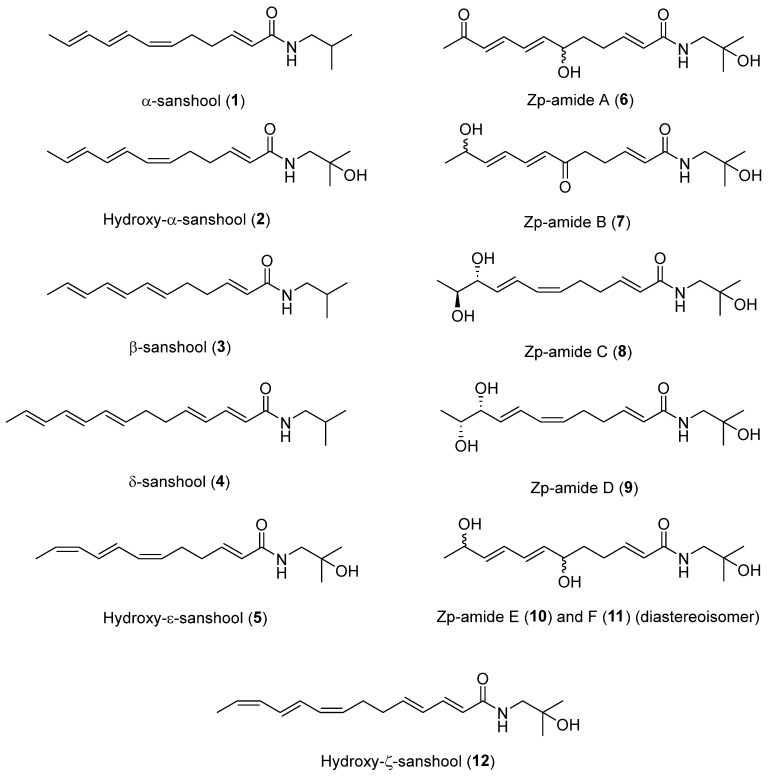
Chemical structures of alkamides in *Z. piperitum*.

**Figure 2 ijms-25-12228-f002:**
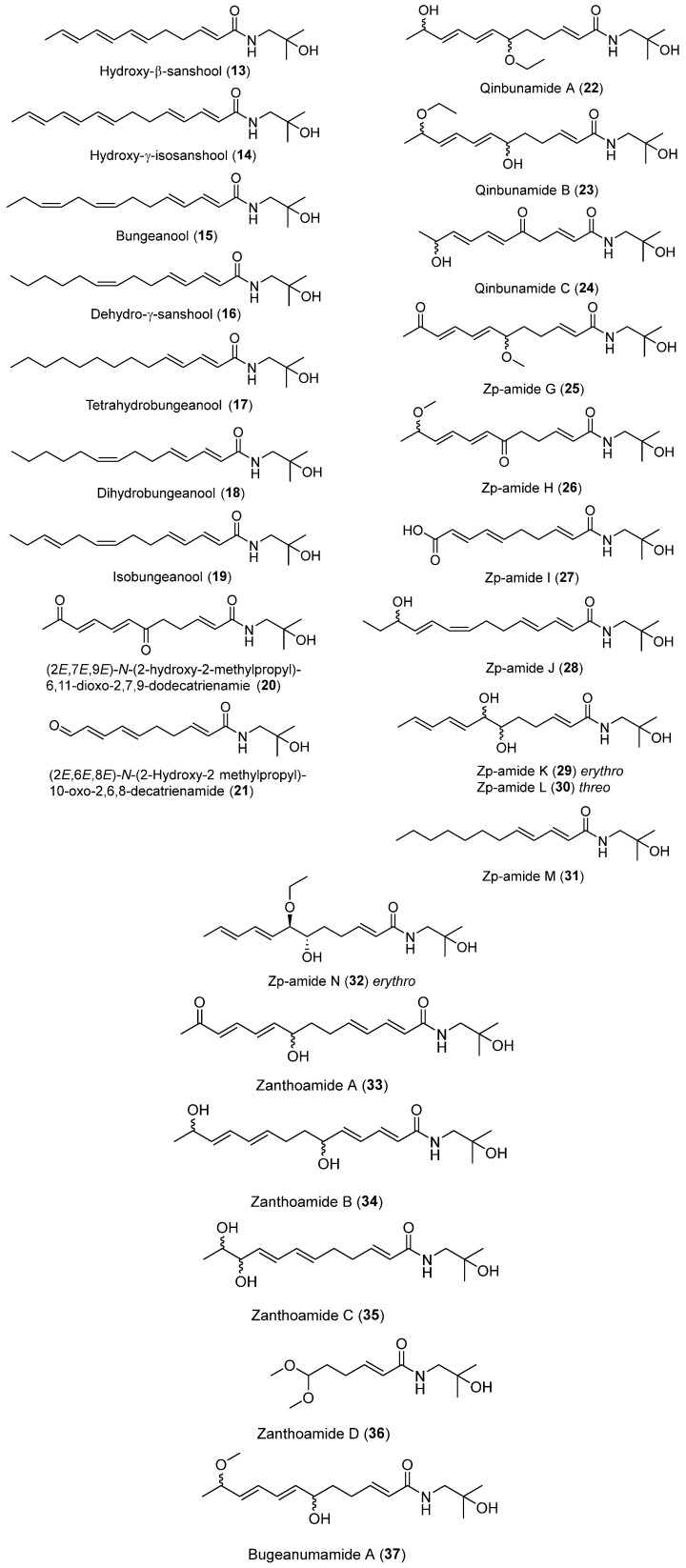
Chemical structures of alkamides in *Z. bungeanum*.

**Figure 3 ijms-25-12228-f003:**
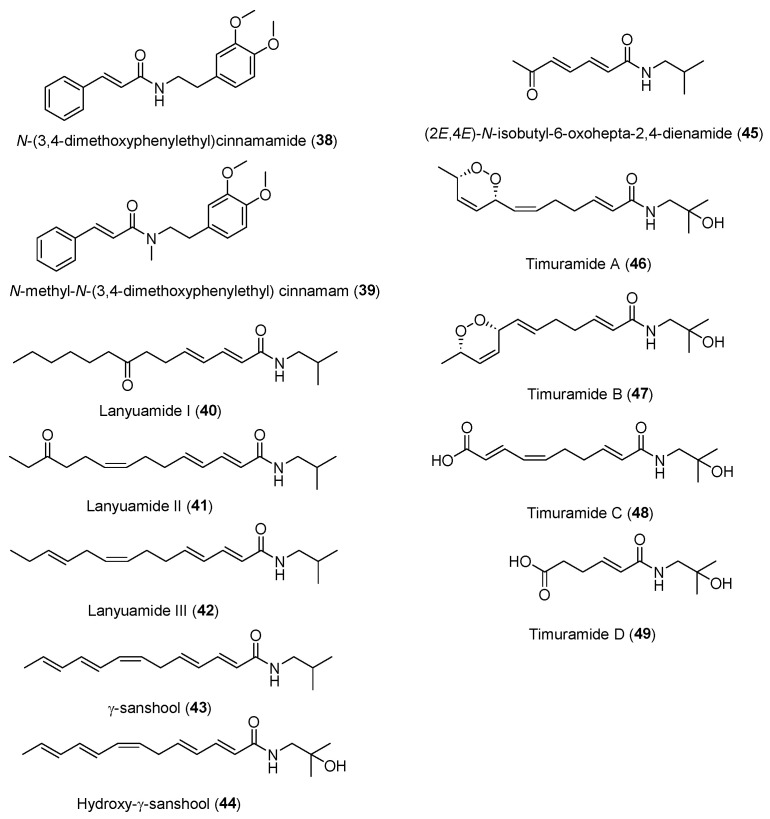
Chemical structures of alkamides **38** and **39** in *Z. lemairie*, **40**–**42** in *Z. Integrifoliolum*, **43**–**45** in *Z. ailanthoides* and **46**–**49** in *Z. armatum*.

**Figure 4 ijms-25-12228-f004:**
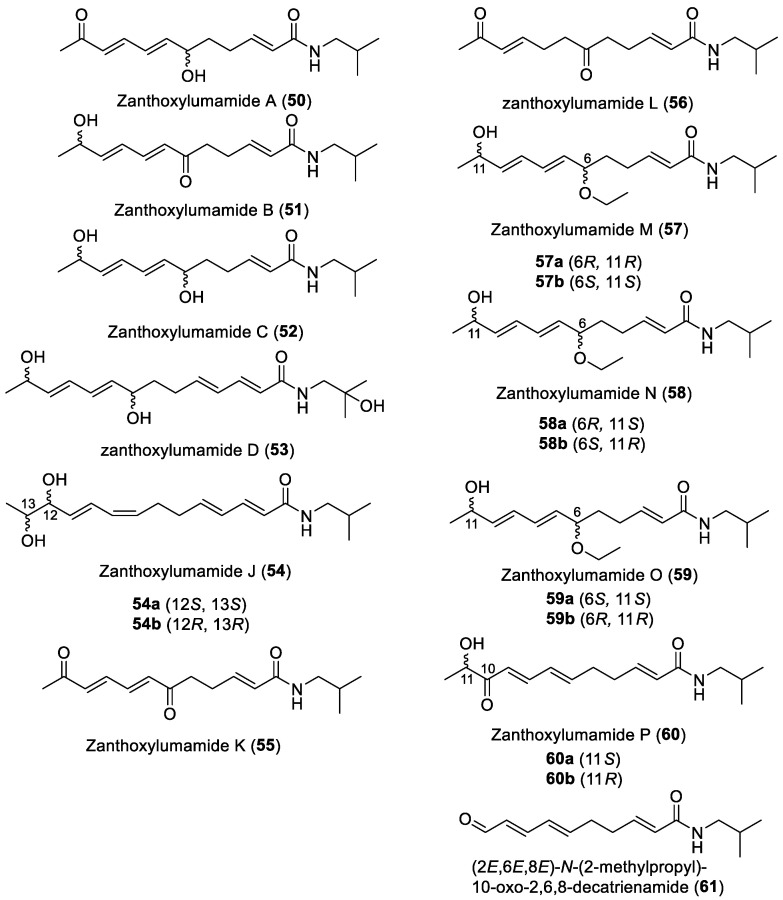
Chemical structures of alkamides **50**–**61** in *Z. nitidum* (Roxb.) DC., **62**–**66** in *Z. nitidum* var. tomentosum, **67** and **68** in *Z. heitzii*, **69** and **70** in *Z. zanthoxyloides* and **71** in *Z. chalybeum*.

**Figure 5 ijms-25-12228-f005:**
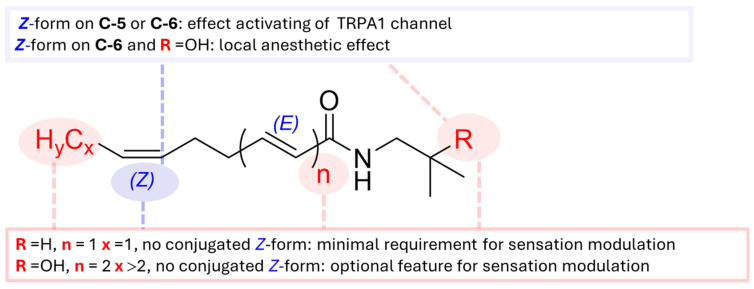
Structural characteristics of alkamides that influence sensory properties of modulation, analgesic effect and local anesthetic effect.

**Table 1 ijms-25-12228-t001:** Alkamides from *Zanthoxylum* species with dysfunctional sensory properties, analgesic or local anesthetic effect.

Alkamides	Sensory Properties, Analgesic and Local Anesthetic Effect	References
α-sanshool (1) 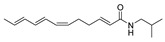	The pungent qualities: burning, tingling and numbing. Burning and tingling were predominantly perceived and lasting longer than numbness.	[84]
Hydroxy-α-sanshool (HAS) (2) 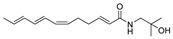	Numbness: during treatment of toothache. Inducing: insensitive to innocuous thermal or tactile stimuli; insensitive to touch or cooling.	[85]
	Analgesia: during treatment of toothache. Tingling and numbing.	[84]
	Promoting Ca^2+^ influx in cells transfected with TRPV1 and evoked robust inward currents in cells transfected with TRPV1 in dorsal root ganglia neurons and trigeminal ganglion neurons.	[86]
	Exciting sensory neurons by inhibiting two-pore-domain K+ channels (KCNK3, KCNK9 and KCNK18).	[87]
	Inhibiting the activity of multiple voltage-gated sodium channel subtypes, among which Nav1.7 is the most strongly affected. Inhibiting Aδ mechanonociceptors that mediate both sharp acute and inflammatory pain.	[88]
	HAS + nanostructured lipid carriers had excellent anesthetic effect at low dose in formalin test compared with free HAS only and lidocaine: worked rapidly and sustained longer effect time.	[89]
β-sanshool (3) 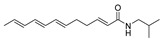	The pungent qualities: numbing and bitter.	[84]
Hydroxy-β-sanshool (13) 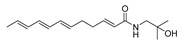	The pungent qualities: numbing, astringent and bitter. Mediating numbing and anesthetic effect.	[61]
γ-sanshool (43) 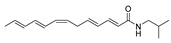	The pungent qualities: burning, numbing, fresh, bitter.	[84]
	Burning and fresh. Potent agonist of TRPV1 activity that explains its pungent and tingling sensation and its use as a natural anesthetic for toothache.	[84]
Hydroxy-γ-isosanshool (14) 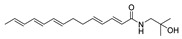	Inducing numbing and anesthetic effect.	[61]
Hydroxy-ε-sanshool (5) 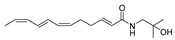	Tingling sensations when applied to the human tongue.	[85]

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
