# Peer review of "Alkamides in Zanthoxylum Species: Phytochemical Profiles and Local Anesthetic Activities"

_ijms, 2024, doi:10.3390/ijms252212228_

Round 1
Reviewer 1 Report
Comments and Suggestions for Authors
Comments:
Lu et. al. comprehensively reviewed alkamides in their phytochemical profiles and sensory effects. The isolation and characterization were well-summarized and presented. The pharmacological and pharmacokinetic properties and clinical results were discussed in-depth. Overall it's recommended to be published with revisions.
Suggestions:
1. Please double-check Table 1. All structures in the left column are clipped off. So please make sure that the structures are correctly sized.
2. Line 240: "by 1D and 2D NMR correlation of C-4 to C-7“. Can you confirm the 2D NMR of the correlation of C-4 to C-7? It's very unusual to observe a C-C correlation, though not impossible. But it's more common to observe H-C HMBC correlation.
3. Line 375-376: "Assign E/Z configurations to the double bonds that often appear in the acyl chains of alkamides from plants." It's difficult to understand what this sentence means, as it seems to be out of context with the rest of the paragraph. Please rephrase here.
4. Line 391: "Based on the above results, E/Z isomerism is not the only element for pungent and tingling properties.” Please rephrase this sentence. The previous paragraph only discussed the effect of E/Z configuration on the sensory properties. Why does this paragraph start with this statement?
Comments on the Quality of English Language
The quality of English can be improved to enhance the quality of the manuscript. Please see the comments in the suggestions.
Reviewer 2 Report
Comments and Suggestions for Authors
The sentence “Resultantly, modulation of sensory nerves by key constituents enriched in herbs should be further evaluated for their local anesthetic effects” does not make sense. It has to be deleted.
Can the authors explain what they are trying to convey through these sentences?
“Resultantly, modulation of sensory nerves by key constituents enriched in herbs should be further evaluated for their local anesthetic effects. Sensory modulation through key factors is also critical for achieving local anesthesia. For instance, the local anesthetic effect induced in sensory neurons is mediated by the capsaicin receptor and transient receptor potential (TRP) channels [83]. In traditional herbal medicines, alkamides in Z. species exhibit sensory modulation like suppressing pain [8, 9]. These pharmacological properties have the potential to develop and identify critical constituents with local anesthetic effects”. “In this section, alkamides in Z. species will demonstrate pharmacological properties ranging from pungent and tingling sensations to analgesic effects and local anesthesia; modulations of critical factors related to sensation or sensory nerves by alkamides will be described”
Under analgesic effect (3.2), authors describe the influx of calcium, a pungent sensation, etc., How are these related to the analgesic effect? Can the authors elaborate on these?
The authors have mentioned the local analgesic effect under the Local anesthetic effect (3.3). Are anesthesia and analgesia the same? What is the local analgesic effect??
Comments on the Quality of English Language
The information has been repeated several times. Authors need to carefully revise the manuscript to avoid giving the same information
Several typographical errors need to be corrected
Round 2
Reviewer 2 Report
Comments and Suggestions for Authors
The manuscript has been revised and it can be accepted for publication